# Relevance of the Preparation of the Target for PLD on the Magnetic Properties of Films of Iron-Doped Indium Oxide

**Hasan B. Albargi [1,2], Marzook S. Alshammari [3,*], Kadi Y. Museery [3], Steve M. Heald [4], Feng-Xian Jiang [5], Ahmad M. A. Saeedi [1], A. Mark Fox [1] and Gillian A. Gehring [1,*]**

[1] Department of Physics and Astronomy, Hicks Building, The University of Sheffield, Sheffield S3 7RH, UK; albargih@yahoo.com (H.B.A.); asaeedi1@sheffield.ac.uk (A.M.A.S.); mark.fox@sheffield.ac.uk (A.M.F.)

[2] Department of Physics, Najran University, P.O. Box 1988, Najran 11001, Saudi Arabia

[3] The National Centre for Laser and Optoelectronics, King Abdulaziz City for Science and Technology, KACST P.O. Box 6086, Riyadh 11442, Saudi Arabia; kmuseery@kacst.edu.sa

[4] Advanced Photon Source, Argonne National Laboratory, Argonne, IL 60439, USA; heald@aps.anl.gov

[5] School of Chemistry and Materials Science of Shanxi Normal University & Key Laboratory of Magnetic Molecules and Magnetic Information Materials of the Ministry of Education, Linfen 041004, China; jfx9902@163.com

* Correspondence: alshammari@kacst.edu.sa (M.S.A.); g.gehring@sheffield.ac.uk (G.A.G.)

**Abstract:** This paper concerns the importance of the preparation of the targets that may be used for pulsed laser deposition of iron-doped indium oxide films. Targets with a fixed concentration of iron are fabricated from indium oxide and iron metal or one of the oxides of iron, FeO, $Fe_3O_4$ and $Fe_2O_3$. Films from each target were ablated onto sapphire substrates at the same temperature under different oxygen pressures such that the thickness of the films was kept approximately constant. The films were studied using X-ray diffraction, X-ray absorption (both XANES and EXAFS), optical absorption and magnetic circular dichroism. The magnetic properties were measured with a SQUID magnetometer. At the lowest oxygen pressure, there was evidence that some of the iron ions in the films were in the state $Fe^{2+}$, rather than $Fe^{3+}$, and there was also a little metallic iron; these properties were accompanied by a substantial magnetisation. As the amount of the oxygen was increased, the number of defect phases and the saturation magnetisation was reduced and the band gap increased. In each case, we found that the amount of the oxygen that had been included in the target from the precursor added to the effect of adding oxygen in the deposition chamber. It was concluded that the amount of oxygen in the target due to the precursor was an important consideration but not a defining factor in the quality of the films.

**Keywords:** PLD; target preparation; room temperature ferromagnetism; dilute magnetic semiconductor; Indium oxide; $(InFe)_2O_3$

## 1. Introduction to the Growth of Oxide Films Using Pulsed Laser Deposition

There is a great interest in the magnetic properties of thin oxide films for use in sensors. Pulsed laser deposition (PLD) is one of the most commonly used growth techniques [1,2]. Particular examples are pure and doped $In_2O_3$, ZnO and cuprate superconductors. A common feature of these oxides is that their magnetic and electrical properties depend strongly on the amount of oxygen that is incorporated into the film and the grain size [3–5]. PLD is a particularly versatile technique because the oxygen stoichiometry can be controlled by depositing the film in a chamber that contains some oxygen gas,

and the grain size and quality of the films depend on the substrate temperature [5–7]. Almost all groups have used targets made using conventional solid-state reaction techniques to fabricate their targets.

An exception to this showed that good quality PLD films of oxides of indium could be made using a metallic target and using the oxygen pressure in the chamber to obtain an oxide film. This was done using targets that contained both metallic indium and tin that were ablated in an oxygen pressure of 7.5 Torr and a silicon substrate heated to 500 °C to make an NO gas sensor [8]. Films of CdO doped with indium where the target was $Cd_{1-x}In_x$ with $x = 0.049$ were ablated in an oxygen atmosphere of 75 mTorr onto a quartz glass substrate held at 300 °C to fabricate CdO which is a good transparent conductor [9].

Growth using an oxide target involves grinding and sintering powders and then pressing them into a target and finally sintering again. There are several variables here, including how the grinding was performed, how many times the powders were ground and sintered and the highest temperature used to sinter the target. In many publications, these details are not given. An interesting comparison was made between the properties of PLD-ablated films of $(InFe)_2O_3$ formed when the grinding was done mechanically compared with using hand grinding using a pestle and mortar. The Fe ions that were in the films that had been ablated from a target that had been formed using mechanical grinding were almost entirely present as a secondary phase of $Fe_3O_4$ [10]. Essentially all the Fe ions were on In sites in films made using the exact same protocol of grinding and sintering but using hand grinding with a pestle and mortar [5].

Another relevant factor is the maximum temperature used to anneal the target. Good films of pure and calcium-doped yttrium iron garnet (YIG) were deposited using a target that had been made from $Fe_2O_3$, $Y_2O_3$ and, where appropriate, CaO. In this case, the target had to be annealed at a high temperature, 1200 °C, for 15 h because targets sintered at lower temperatures were brittle and were destroyed during laser ablation [11,12]. Good quality films were grown on a gadolinium gallium garnet (GGG) using a substrate temperature of only 500 °C after the films had subsequently been annealed in air at 1000 °C [11,12].

The annealing temperature for targets of pure $Fe_2O_3$ was important for a different reason. In this case, the target retained its orange colour when it was annealed at 500 °C and then could be used to grow films of maghemite by PLD at 100 mTorr. However, if the target had been annealed at 1200 °C in air it changed its colour to black and then could be used to get films of $Fe_3O_4$ and FeO [12].

Another situation where a target changed colour after annealing to a high temperature occurred with ZnCoO. A target was made from metallic cobalt and ZnO and was ground and sintered repeatedly. It retained its light grey colour when the maximum temperature used for annealing was 1000 °C but the colour changed to dark green if it was annealed at a higher temperature ~1150 °C. The ordered compound of $Zn_{1-x}Co_xO$ is green (Rinman's Green) hence it was clear that in this case high temperatures are required to complete the solid-state reaction. Films that contained 10% cobalt and were of similar thickness were ablated from the targets annealed at 1000 and 1150 °C and were compared. It was found that the film made from the target with the 1000 °C anneal had a significant content of metallic cobalt present as nanoparticles that caused blocking behaviour at 30 K. The film made with the target that had been annealed at 1150 °C had a much larger saturation magnetisation and any nanoparticles of cobalt were too small to show blocking behaviour above 5 K [13].

Tuning the oxygen content in the films of pure and doped ZnO by changing the oxygen pressure in the chamber has been performed very widely. However, the oxygen content of the target can also be controlled by the precursor. To investigate the relevance of this, a study was made of films of ZnCoO using three different precursors in the targets: Metallic cobalt, CoO and $Co_3O_4$ [14]. Most previous work had used $Co_3O_4$. This study demonstrated that in this case, these different precursors produced different films even though there was no trace of the precursors in the ablated films. It also showed that using metallic cobalt as a precursor had effects beyond the concentration of oxygen and that the subtle chemistry of PLD was also affected [14].

In$_2$O$_3$ is an *n*-type transparent semiconductor material with a wide band gap of 3.75 eV that is in the ultraviolet (UV) region of the spectrum [15,16]. This material is an insulator in its stoichiometric form, while in its oxygen deficient form it has *n*-type doping levels that are induced by oxygen vacancies. The stoichiometry is an important factor in determining electrical properties [17,18]. In$_2$O$_3$ can grow in three different structures; however, all thin films that were grown by PLD grow in the cubic bixbyite structure as is seen here. In this structure, each cubic unit cell of In$_2$O$_3$ contains 16 formula units (80 atoms) and has a lattice constant of 10.118 Å [19,20].

In$_2$O$_3$ has been doped with transition metals, in particular, iron, to form $(In_{1-x}Fe_x)_2O_3$ and in this case the lattice constant decreases monotonically with increasing Fe concentration until $x = 0.2$, indicating that the maximum solubility limit of Fe ions in In$_2$O$_3$ lattice is approximately 20%. The saturation magnetisation, $M_s$, has also been found to increase proportionally with increasing Fe concentration, for $x$ between 0.05 and 0.2 [15,21–25].

It is generally found that oxygen vacancies induce defects that are responsible for the ferromagnetism and that magnetism occurs when the Fe ions are reduced from $Fe^{3+}$ to $Fe^{2+}$ [15,26]. In addition, the value of $M_s$ in $(In_{0.95}Fe_{0.05})O_3$ thin films has been found to be affected by the grain size, where the highest magnetisation saturation corresponds to the largest grain size implying that grain boundary magnetism is not important for these films [5].

However, there are also reports of Fe-doped In$_2$O$_3$ thin films that contain magnetic nanoparticles of Fe$_2$O$_3$ or Fe$_3$O$_4$ [20,25,27]. The presence of the Fe$_3$O$_4$ nanoparticles has been reported to enhance the room temperature magnetisation, magnetoresistance and a larger value of the coercive of ~400 Oe [10].

Previous work has considered Fe$_2$O$_3$ to be the obvious precursor to use with In$_2$O$_3$ to generate a target for the PLD because it should generate a stoichiometric target [5,10,15,19,24,26]. It is well known that if a film is ablated from a target in a high vacuum, the film will contain less oxygen than the target because some oxygen is lost in the PLD process. Hence, films are grown in different oxygen pressures so as to control the density of oxygen vacancies. In this work, we describe the effects of controlling the density of oxygen vacancies by changing the precursor used to fabricate the target as well as controlling the amount of oxygen in the growth chamber. We have made PLD films from targets that contain 5% iron using metallic iron, FeO, Fe$_3$O$_4$ and Fe$_2$O$_3$ together with In$_2$O$_3$ in the targets. All other conditions were kept constant.

The films were studied using X-ray diffraction to measure the change in the lattice constant with the changing density of oxygen vacancies and also X-ray absorption, X-ray absorption near edge structure (XANES) and extended X-ray absorption fine structure (EXAFS) techniques to measure the state of ionisation of the Fe and its environment. The hysteresis loops of the films were measured at room temperature and at 5 K. Optical measurements of the absorption and the magnetic circular dichroism (MCD) were also studied. This enables us to investigate the effects on the films of adding oxygen to the target using a different precursor with that of adding oxygen to the PLD chamber. It will also indicate if there are extra chemical effects of using different targets that occur in the PLD process beyond the effects of having a different concentration of oxygen vacancies. Such effects were found in PLD films made of ZnCoO using different compounds of Co in the fabrication of the target [14].

## 2. Fabrication of the Targets and Growth of the Films

The targets were made using a solid-state reaction method that was performed using the following protocol which we have found to be an effective method to produce targets that could be used to grow good quality films [14]. Appropriate weights of one of the precursors, FeO, Fe$_2$O$_3$ or Fe$_3$O$_4$ and In$_2$O$_3$ chosen so as to give a ratio of 0.05:0.95 of Fe to In, were mixed together; the amounts used and the necessary information required to obtain these values are given in Table 1. The powders were purchased from Alfa Aesar (Karlsruhe, Germany) and had purities of 99.999% for In$_2$O$_3$, 99.995% for FeO, 99.998% for Fe$_3$O$_4$ and 99.999% for Fe$_2$O$_3$. The powders were hand ground for 30 min in a ceramic pestle and mortar and calcined in air at 300 °C for 12 h. They were then ground again for a further 30 min and sintered in air at 600 °C for 12 h. The procedure was repeated with the sintering

temperature raised to 900 °C. After the final anneal, the mixture was placed in a *Specac (*Specac Ltd., Kent, England) die, which was evacuated with a roughing-pump and, using a manual hydraulic press, compressed to 25000 kPa. This produced a relatively dense, cylindrical pellet of diameter 25 mm and thickness between 2 and 5 mm, depending upon the amount of the initial powders used. The pellet was then given a final anneal at a maximum temperature of 1000 °C.

**Table 1.** The weight of each material required to make targets weighing ~11 g.

| Precursor | Molecular Weight | Element | Number Per Gram $\times 10^{21}$ | Number of Grams for a Target of $(In_{1-x}Fe_x)_2O_3$ |
|---|---|---|---|---|
| $In_2O_3$ | 277.64 | In | 4.3381 | 10 |
| Fe | 55.845 | Fe | 10.784 | 0.212 |
| FeO | 71.844 | Fe | 8.3822 | 0.272 |
| $Fe_3O_4$ | 231.533 | Fe | 7.8030 | 0.292 |
| $Fe_2O_3$ | 159.69 | Fe | 7.5422 | 0.303 |

Thin films of thickness of approximately 200 nm were deposited on double-side polished sapphire c-cut $Al_2O_3$ (0001) substrates that were held at 450 °C. The deposition used a Lambda Physik LEXTRA 200 XeCl excimer laser (Lambda Physik Lasertechnik, Goettingen, Germany) with a maximum power of 400 mJ per pulse, an operating wavelength of 308 nm, and a 10 Hz repetition rate of 28 ns pulses. The target was rotated at 60 rpm and was placed 5 cm from the substrate. We had previously checked that there was almost no difference in the films that were made using the XeCl laser compared with the, more standard, KrF laser. Three films were made from each target at each of three different oxygen pressures in the PLD chamber. The three conditions were base pressure, $2 \times 10^{-5}$ Torr and oxygen pressures of $2 \times 10^{-4}$ Torr and $2 \times 10^{-3}$ Torr. This was to allow us to compare the effects of adding oxygen to the target from the precursor with that of adding oxygen to the PLD growth chamber.

## 3. Results

### 3.1. Structural Characterisation of the Films

The films' structural and chemical characteristics were obtained using X-ray diffraction XRD, (Rigaku Corporation, Tokyo, Japan and Bruker D2 Phaser, Coventry, UK), XANES and EXAFS techniques. These techniques gave us information on the lattice constant and grain size of the $In_2O_3$ matrix and the presence of any nanoparticles that existed in the films.

The XRD data, shown in Figure 1, were measured using Cu K$\alpha$ radiation ($\lambda$ = 1.5406 Å) using a $\theta-2\theta$ scan. For the samples grown at base pressure, the data showed that the samples had diffraction peaks corresponding to (222) and (400) of the pure cubic bixbyite $In_2O_3$; the (006) peak is from the sapphire substrate. A small peak at ~36° (shown in red) indicated the presence of the secondary phase of FeO; however, no peaks from metallic iron were detected. The insets in Figure 1 show an enlarged plot of the (222) reflection; at base pressure, all three lattice constants were 10.18 ± 0.02 Å but there are real differences for the films grown at $2 \times 10^{-3}$ Torr where the lattice constants for FeO, $Fe_3O_4$ and $Fe_2O_3$ were 10.17 ± 0.02 Å, 10.14 ± 0.02 Å and 10.12 ± 0.02 Å, respectively. This is in agreement with earlier results where it was found that the lattice constant increased slowly with increasing oxygen due to the elimination of isolated oxygen vacancies, but that at higher oxygen pressure it decreased rapidly due to the removal of the oxygen vacancy being accompanied by the oxidation of the large $Fe^{2+}$ ion to the much smaller $Fe^{3+}$ ion [15,26]. The size of the observed lattice contraction increased with the amount of oxygen in the target. Hence, the data indicated that the total amount of oxygen in the films depends on both the oxygen in the precursor as well as the oxygen in the PLD chamber.

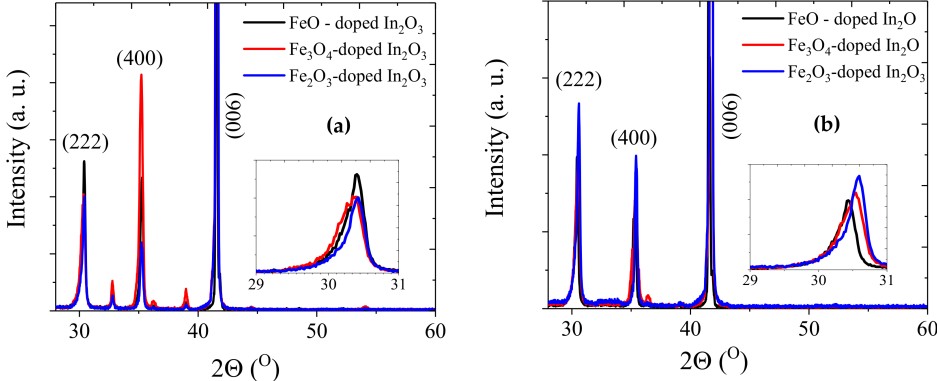

**Figure 1.** XRD data of the Fe-doped $In_2O_3$ thin films grown from different oxide precursors at (**a**) base pressure of $2 \times 10^{-5}$ Torr and (**b**) $2 \times 10^{-3}$ Torr. The insets demonstrate the effect of the precursor on the position of the (222) peak.

To have a more accurate estimate of the presence of defect phases and the ionisation state and environment of the Fe ion, K-edge XANES and EXAFS spectra have been measured. The films grown at base pressure and the powder oxide standards were measured on beamline 20-BM, and the films grown at higher oxygen pressure were measured on beamline 20-ID at the Advanced Photon Source. The setups on both beamlines were similar with Si (111) monochromators providing 1 eV energy resolution at the Fe K edge. The measurements were made at room temperature at a glancing angle of ~5° with the X-ray polarization normal to the surface of the films. Multielement solid-state detectors (4 element silicon drift detector on 20-ID and 13 element Ge detector on 20-BM) were used for fluorescence detection, and the samples were spun at a few Hz to avoid Bragg reflection interference from the single-crystal substrates. Typically, 4–8 scans were averaged for improved signal to noise. Data were analysed using the Demeter analysis package [28].

In the XANES spectra, as shown in Figure 2, the signals from wüstite (FeO), magnetite ($Fe_3O_4$) and hematite ($Fe_2O_3$) have been plotted alongside Fe-doped $In_2O_3$ films to be used as references. Figure 2a displays the data from the samples grown at base pressure and indicates an absorption at ~7117 eV (marked with an arrow) that is at a lower energy than for wüstite. This means that they all contain a small percentage of metallic iron. Such results are caused by the increased number of oxygen vacancies that are generated by PLD at base pressure [29] and all evidence of metallic iron has vanished from the films made at $2 \times 10^{-4}$ Torr and $2 \times 10^{-3}$ Torr. The XANES data for the films deposited at higher oxygen pressure, $2 \times 10^{-3}$ Torr, are close to that of hematite, $Fe_2O_3$, indicating that most of the Fe ions are in the state $Fe^{3+}$ although a small fraction of the Fe ions may be present as $Fe^{2+}$.

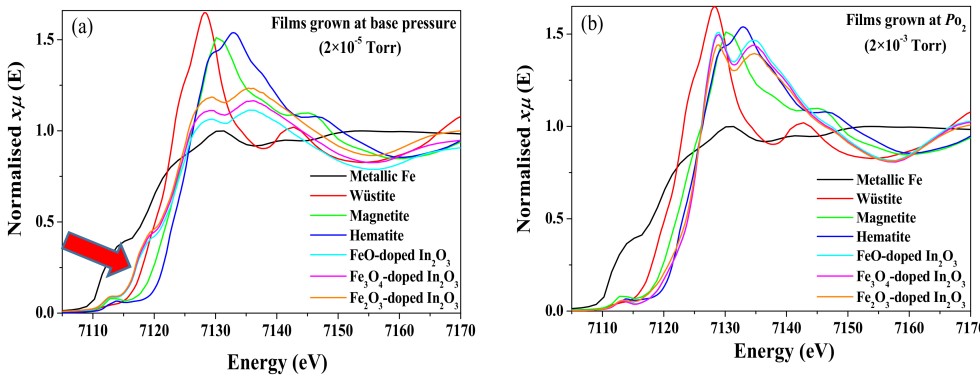

**Figure 2.** K-edge X-ray absorption near edge structure (XANES) spectra of reference compounds of metallic Fe, FeO, $Fe_3O_4$ and $Fe_2O_3$ and the Fe-doped $In_2O_3$ films grown from different precursors at (**a**) at base pressure of $2 \times 10^{-5}$ Torr; where an arrow at 7117 eV indicates an additional absorption and (**b**) higher partial oxygen pressure of $2 \times 10^{-3}$ Torr.

The environment of the Fe ions is obtained from an analysis of the Fourier transform of the EXAFS and the results are shown in Figure 3. The results for the films grown at base pressure are shown in Figure 3a and are compared with the EXAFS spectrum from a sample of $Fe_2O_3$-doped $In_2O_3$ sample that is believed to be pure substitutional [5]. All of the data shows a strong peak near $R = 1.6$ Å from near neighbour oxygens. Note that the transforms are not phase corrected, so the peaks are shifted ~0.3 Å lower from their actual distances. The spectra from the films that were grown at base pressure have similar peaks in the region $R = 2$–4 Å as the fully substitutional spectrum (shown in red) except near $R$~2.1 Å where they show small additional peaks that are likely characteristic of Fe metal. If the metallic fraction is very small, it is difficult to detect it in diffraction. Attempts at fitting with a metal site combined with a substitutional site were only moderately successful. A good fit was achieved with about 12% of the doped Fe in the metallic environment, but the substitutional site parameters had to be modified more than seems reasonable to accommodate the Fe. This indicates the possibility of a third type of Fe oxide site, in agreement with the XRD where a small signal from FeO was also detected. Unfortunately, the data range was such that it is difficult to reliably fit a three-site model. The reduction in the intensity for $2 \leq R \leq 3$ Å, also suggests the presence of some Fe oxide secondary phase in addition to the metallic Fe clusters.

Detection of an impurity phase by XRD depends on the ability to measure the square of the concentration. In this case, this is given by the percentage of Fe in the whole sample, 5%, and the percentage of these atoms that are in a metallic environment, about 12%. These combine to give a percentage of metallic Fe in the sample that is about 0.6% which was not measurable.

The data shown in Figure 3b are from the samples grown at $2 \times 10^{-3}$ Torr. A plot of a reference sample of hematite has been included because the XANES data from the films had indicated that all the iron was in the $Fe^{3+}$ state which is also characteristic of hematite. These data show that the atomic arrangement in the films is very close to that expected for Fe ions substituted on the In sites and distinct from that of hematite. At higher oxygen pressures, the Fe ions appear to be substitutional for all the targets and show no evidence of any metallic iron or any iron oxide. There are some differences in the edge data between our sample and the reference sample that may be due to the result of better structural order in our samples. While the basic structure of the peaks looks similar, there is an increasing difference in the height of the peaks at larger distances that could be a characteristic of disorder. The $In_2O_3$ has two different In sites and the disorder arises if the Fe substitutes randomly on both sites [15,27].

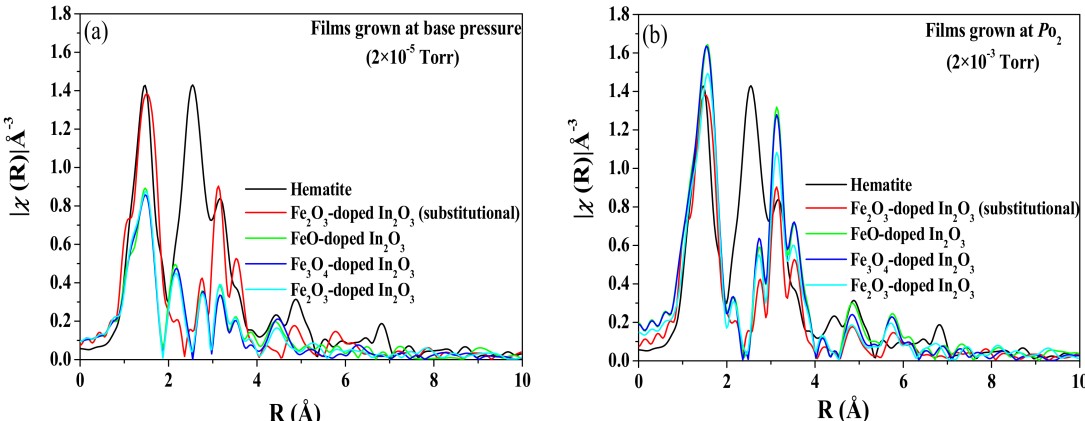

**Figure 3.** X-ray absorption fine structure (EXAFS) Fourier transform of $Fe_2O_3$ and substitutional $Fe_2O_3$-doped $In_2O_3$ deposited at base pressure as reference compounds and the Fe-doped $In_2O_3$ films grown from different precursors at (**a**) base pressure of $2 \times 10^{-5}$ Torr and (**b**) $O_2$ pressure of $2 \times 10^{-3}$ Torr.

In summary, we find evidence of defect phases, metallic iron from the XANES and FeO from the XRD and the EXAFS in films grown at base pressure. The results of the XANES and the EXAFS

measurements of films grown with oxygen in the chamber indicated that substantially all the $Fe^{3+}$ ions were situated on $In^{3+}$ sites.

### 3.2. Magnetic Properties of the Films

Magnetic hysteresis loops were taken at 5 and 300 K for the substrates and all the Fe-doped $In_2O_3$ films using a Quantum Design SQUID magnetometer (San Diego, CA, USA). It was found that all the films displayed room temperature ferromagnetism; examples of the loops obtained are shown in Figure 4 and the values of $M_s$ and $H_c$ measured at room temperature for the different precursors and oxygen pressure in the PLD chamber are summarised in Table 2. A magnetic field of 10,000 Oe was applied parallel to the plane of the film during the magnetisation measurements. The diamagnetic contribution from the sapphire substrate was subtracted, as was the paramagnetic contribution from the film from the data shown in Figure 4.

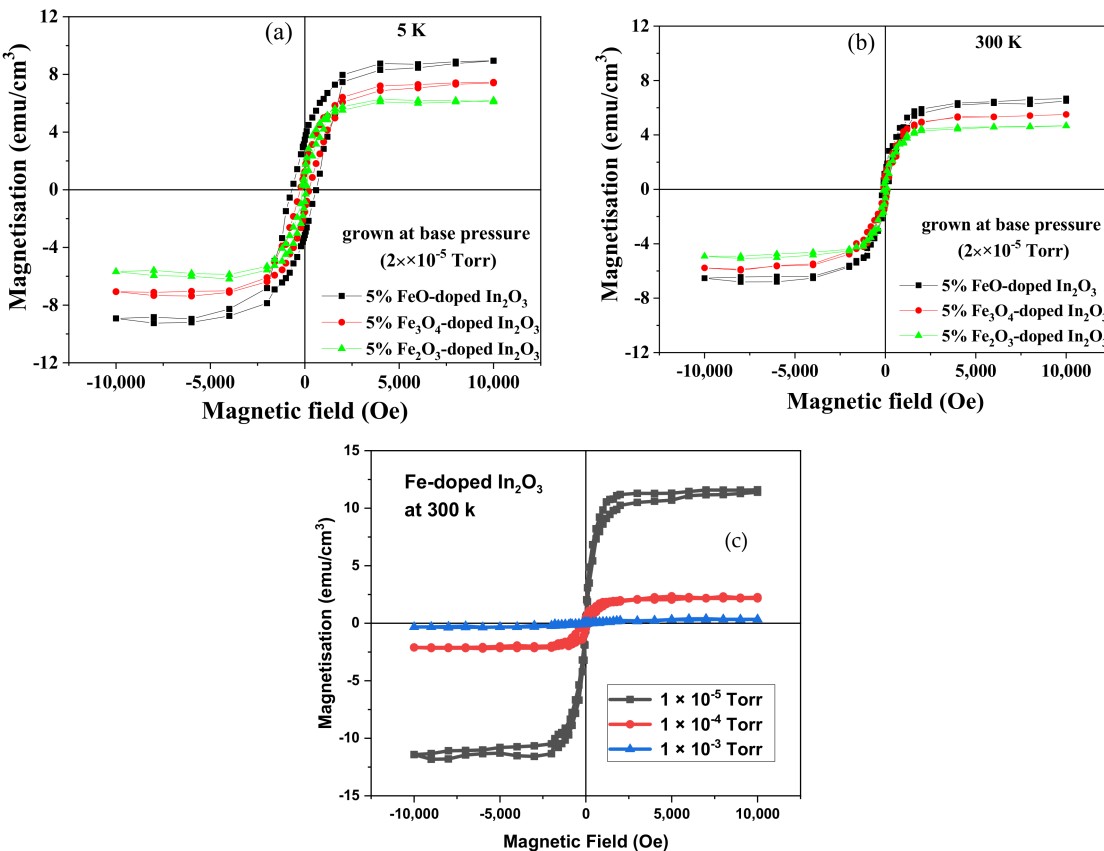

**Figure 4.** (**a**) and (**b**) are respectively the magnetic hysteresis loops measurements at 5 and 300 K for the Fe-doped $In_2O_3$ films from different oxide precursors at a base pressure of $2 \times 10^{-5}$ Torr, (**c**) shows the room temperature data for films that were made with Fe metal in the precursor. The diamagnetic and paramagnetic terms have been subtracted from all the data shown here.

The paramagnetic contribution from the film, seen at 5 K, was consistent with it being due to free spins of all the $Fe^{3+}$ ions because the value of $p_{eff}$ was found to be 4.5 ± 0.3 compared with the expected value for $Fe^{3+}$ of 4.9 [30]. The observed strong temperature dependence of the saturation magnetisation was observed previously in films of Fe-doped $In_2O_3$ that had semiconducting behaviour [3,15,31]. Part of the magnetisation observed from the films deposited at base pressure could be due to the 12% of the Fe ions that were in a metallic environment as observed by EXAFS leading to a metallic concentration of about 0.6%. If each of these contributed 2.2 $\mu_B$ to the bulk magnetisation, this would contribute about 1.3 emu/cm$^3$ to the magnetisation. This is comparable to the difference between the

magnetisation observed for films grown at base pressure and those grown at $2 \times 10^{-4}$ Torr for the films grown with the oxide precursors.

Previous results had found that oxygen vacancies and $Fe^{2+}$ ions were necessary for ferromagnetism to be observed in $Fe\text{-}In_2O_3$ and these results are consistent with this earlier work [3,31,32]. This pattern was seen for all films and all oxygen content because it was found that both the saturation magnetisation and coercive field decreased with oxygen content whether the oxygen was in the target or in the PLD chamber. The decrease in magnetisation for added oxygen was larger for the films deposited at base pressure. The large value of $M_s$, seen only at base pressure, may be due to the existence of metallic iron as seen by XANES as well as a larger number of oxygen vacancies [3,31,32]. The coercive field of the films deposited at base pressure from the FeO target was increased significantly at 5 K compared with that measured at 300 K, leading to the deduction that blocked magnetic nanoparticles, of probably Fe metal, existed at low temperatures in that film. This increase was not seen for the films deposited at higher pressure. The magnetisation of the films made from the target that contained metallic iron as the precursor showed a stronger dependence on oxygen content than the films made from the oxide precursors. Interestingly, the coercive field seen at 5 K for the film grown from the Fe metal target at base pressure was higher than with those grown from the oxide targets grown at base pressure. This implies that if, as expected, the film grown with a Fe-target does contain a larger percentage of metallic iron, the nanoparticles are so small that their blocking temperatures are close to 5 K, or below, whereas significant blocking of nanoparticles has occurred for the film ablated from the target that had been made with FeO.

Temperature-dependent plots of the magnetisation were measured under the conditions of zero field cooled (ZFC) and field cooled (FC) to investigate further the relative importance of nanoparticles. This magnetisation was obtained for all the samples grown from FeO, $Fe_3O_4$ and $Fe_2O_3$ precursors and deposited at base pressure; this was the condition where the XANES measurement had indicated the presence of about 12% of the iron atoms in a metallic environment. A magnetic field of 100 Oe was applied in parallel to the plane of the samples [33–35]. The diamagnetic contribution from the sapphire substrate was subtracted from all ZFC and FC curves shown in Figure 5. The separation of the ZFC and the FC curves is due to the increase of the anisotropy field to become comparable or larger than the measuring field, 100 Oe, as the temperature is reduced. If the magnetisation had been dominated by nanoparticles, the curves should vary as $1/T$ in the reversible regime, but this is not observed here.

**Table 2.** The values of $M_s$ and $H_c$ measured at 300 K and at 5 K for the different precursors and oxygen pressure in the pulsed laser deposition (PLD) chamber.

| Precursor | Base Pressure | | | | $2 \times 10^{-4}$ Torr | | | | $2 \times 10^{-3}$ Torr | | | |
|---|---|---|---|---|---|---|---|---|---|---|---|---|
| | **Fe** | **FeO** | **Fe$_3$O$_4$** | **Fe$_2$O$_3$** | **Fe** | **FeO** | **Fe$_3$O$_4$** | **Fe$_2$O$_3$** | **Fe** | **FeO** | **Fe$_3$O$_4$** | **Fe$_2$O$_3$** |
| $M_s$ 300 K (5 K) emu/cm$^3$ | 11.0 ± 0.4 (12.0 ± 0.4) | 6.8 ± 0.3 (9.2 ± 0.4) | 6.0 ± 0.2 (7.8 ± 0.3) | 4.2 ± 0.3 (6.1 ± 0.3) | 2.6 ± 0.1 (3 ± 0.2) | 5.1 ± 0.2 (7.3 ± 0.3) | 4.4 ± 0.3 (6.1 ± 0.2) | 3.2 ± 0.2 (5.2 ± 0.2) | 0.3 ± 0.2 (5 ± 0.1) | 3.6 ± 0.3 (4.8 ± 0.2) | 2.5 ± 0.2 (3.9 ± 0.2) | 2.0 ± 0.2 (3.0 ± 0.1) |
| $H_c$ 300 K (5 K) Oe | 25 ± 3 (150 ± 17) | 135 ± 15 (535 ± 32) | 113 ± 1 (271 ± 23) | 100 ± 12 (153 ± 18) | 100 ± 12 (120 ± 16) | 123 ± 14 (127 ± 17) | 106 ± 12 (111 ± 14) | 94 ± 12 (96 ± 14) | 50 ± 7 (80 ± 9) | 118 ± 14 (110 ± 15) | 97 ± 10 (98 ± 14) | 92 ± 10 (87 ± 12) |

The FC/ZFC curves shown in Figure 5a,b for the films made from oxide precursors and grown at base pressure are consistent with the increase in the coercive field at the temperature measured at 5 K as given in Table 1. The large coercive field observed at 5 K for the film grown from the FeO could be due to shape anisotropy of the metallic inclusions but even in this case, the $1/T$ dependence in the reversible region was not observed, indicating that the magnetic contributions from the nanoparticles are not dominating the overall magnetisation.

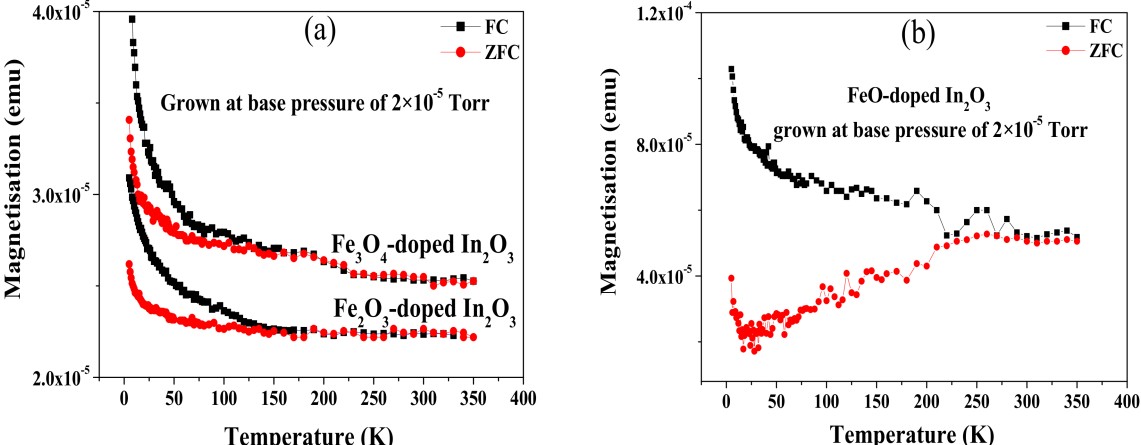

**Figure 5.** Field cooled (FC) and zero field cooled (ZFC) magnetisation curves of the Fe-doped $In_2O_3$ from $Fe_3O_4$ and $Fe_2O_3$ precursors in (**a**) and from FeO precursor in (**b**) where all samples were grown at a base pressure of $2 \times 10^{-5}$ Torr.

The increase of the magnetisation below 50 K is a characteristic of all DMS materials. This behaviour arises from isolated paramagnetic ions which are not contributing to the long-range ferromagnetic order. Existence of these ions was already discussed because they give rise to a paramagnetic contribution to the hysteresis loops [30].

### 3.3. Optical Absorption

The optical properties of the Fe-doped $In_2O_3$ films were investigated by carrying out transmission and reflection measurements at room temperature. From these measurements, absorption data were obtained to gain an insight into the electronic structure and to estimate the density of gap states and the band gap of the films.

The optical properties of this material are sensitive to different targets and film preparation parameters, including the amount of oxygen, the following results will show the effect of changing the oxygen content. Figure 6 illustrates the absorption data at energies close to the band edge for all the Fe-doped $In_2O_3$ films grown at the base and different oxygen pressures. In doped $In_2O_3$ there are two dominant effects that can change the band gap. Isolated oxygen vacancies are donors and will be ionised to increase the band gap due to the Burstein-Moss effect, however, a lattice contraction will increase the band gap. Both effects are relevant here. The values of the band gap are summarised in Table 3. At low oxygen pressure, $2 \times 10^{-5}$ Torr, the films have essentially the same lattice constant and the band gap is highest for the films with the lowest amount of oxygen in the target due to the Burstein-Moss effect. At the higher oxygen pressure, $2 \times 10^{-3}$ Torr, the lattice contraction for the films containing the most oxygen is the dominant effect in determining the lattice constant.

All the spectra show a substantial amount of absorption below the energy gap due to energy states in the gap. We note that the highest density of gap states occurs for the three films grown at base pressure, which were known to contain about 12% of the iron atoms in a metallic environment.

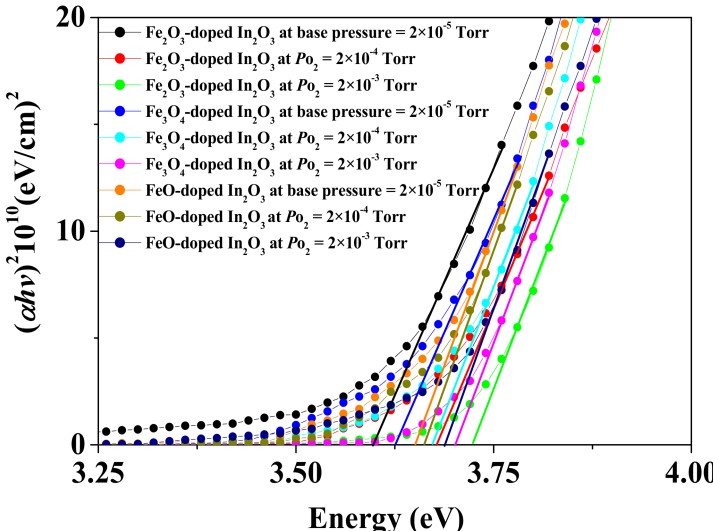

**Figure 6.** Absorption data of Fe-doped $In_2O_3$ samples grown from FeO, $Fe_3O_4$ and $Fe_2O_3$ precursors deposited at a base pressure of $2 \times 10^{-5}$ Torr and the two higher oxygen pressures of $2 \times 10^{-4}$ Torr and $2 \times 10^{-3}$ Torr.

### 3.4. The Magneto-Optical Properties

The MCD spectra for all Fe-doped $In_2O_3$ samples were measured in the energy range between 1.7 and 4 eV at room temperature in Faraday geometry using an applied magnetic field of 18000 Oe, as displayed in Figure 7. The MCD is a very powerful technique because it indicates the amount of spin polarisation that is present in the quantum states that are involved in transitions at a particular energy [36]. The MCD signal from the sapphire substrate has been subtracted from the data shown in Figure 7.

**Table 3.** Summary of the band gap values of Fe-doped $In_2O_3$ thin films deposited at the base and higher oxygen pressures.

| Sample | $E_g$ (eV) | | |
|---|---|---|---|
| | **Base Pressure ($2 \times 10^{-5}$ Torr)** | **Oxygen Pressure ($2 \times 10^{-4}$ Torr)** | **Oxygen Pressure ($2 \times 10^{-3}$ Torr)** |
| FeO-doped $In_2O_3$ | $3.65 \pm 0.01$ | $3.66 \pm 0.01$ | $3.69 \pm 0.02$ |
| $Fe_3O_4$-doped $In_2O_3$ | $3.63 \pm 0.02$ | $3.67 \pm 0.02$ | $3.70 \pm 0.01$ |
| $Fe_2O_3$-doped $In_2O_3$ | $3.60 \pm 0.02$ | $3.68 \pm 0.02$ | $3.72 \pm 0.01$ |

The results taken at base pressure are shown in Figure 7a; these are characteristic of films that contain metallic iron where the MCD may be calculated using the Maxwell–Garnett theory [5,15,37]. The signals indicate that the most metal is in the film made from the FeO target and the least in the one made from the $Fe_2O_3$ target, but in both the percentage of the volume occupied by metal is small, approximately 0.5% or less. In contrast, there is no sign of any metallic iron seen in the MCD spectra of the films made at the higher oxygen pressure shown in Figure 7b, as expected from the XANES results shown in Section 3.1. The MCD spectra varied between positive for the film with the lowest oxygen content, FeO-$In_2O_3$ grown $2 \times 10^{-4}$ Torr, to negative for the film with the largest amount of oxygen, $Fe_2O_3$-$In_2O_3$ grown at $2 \times 10^{-3}$ Torr, as the oxygen content was increased. It had been found earlier that the MCD was positive for films with a high density of carriers produced by oxygen vacancies and small or negative for those in the semiconducting regime [15]. These results are also consistent with the values of the saturation magnetisation obtained from the magnetic hysteresis loops measured by the SQUID shown in Table 1.

The dip observed in all the spectra shown in Figure 7b just above 3.4 eV and below the band gap energy stated in Table 2 is typical of magnetic oxide semiconductors. It arises from transitions from the valance band to empty donor states of oxygen vacancies situated below the conduction band edge and is a clear indication that these donor states are spin polarised [35]. The absorption from these states is seen in Figure 6 in the region of about 0.25 eV below the band gap.

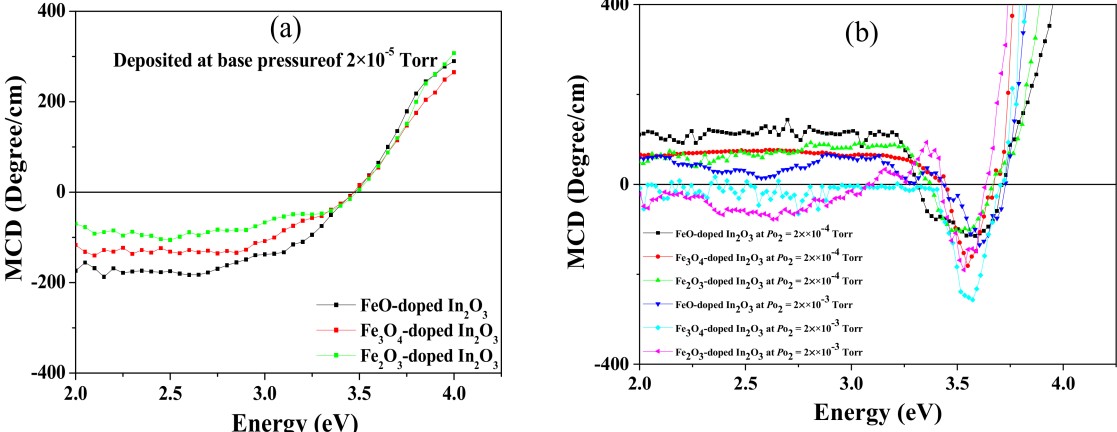

**Figure 7.** Magnetic circular dichroism (MCD) spectral shapes of the Fe-doped $In_2O_3$ samples deposited from different precursors at (**a**) base pressure and (**b**) at $2 \times 10^{-4}$ Torr shown with black, red and green symbols for FeO, $Fe_3O_4$ and $Fe_2O_3$, respectively and at $2 \times 10^{-3}$ Torr with blue, turquoise and pink symbols for FeO, $Fe_3O_4$ and $Fe_2O_3$, respectively.

## 4. Discussion

In this work, we studied a range of films that have differed in two aspects: The amount of oxygen in the PLD chamber and the precursor that was used to add Fe ions to the target material. We anticipated that films that were made with FeO in the target would contain less oxygen than those that were made with $Fe_3O_4$, which in turn would have less oxygen than those made with $Fe_2O_3$, however, we speculated that there may be subtle questions of chemistry of the PLD process that come into play, as we found in Co ions incorporated in ZnO [14].

The aim of this work was to see if the extra oxygen that is incorporated into the target enters the films in a similar way as the oxygen included in the PLD chamber. We investigated this by keeping all other variables constant. All the films had the same percentage of iron (5%), the same procedures for mixing and annealing the targets, the same substrate temperature and, as far as possible, the same thickness of the films. The processing of the targets was done in air, so it was interesting to observe that the amount of oxygen in the target still depended on the precursor and was not equalised during the process of fabrication.

The three precursors have different magnetic properties: FeO and $Fe_2O_3$ are antiferromagnetic, or very weakly ferromagnetic, whereas $Fe_3O_4$ is strongly ferromagnetic. Hence, it was important to check if any of the $Fe_3O_4$ precursor had survived in the films and the XAFS data showed clearly that it had not. Nor was there any suggestion that the magnetic properties of the film made with this precursor were significantly inconsistent with the other films.

We had evidence of defect phases of both metallic iron and FeO appearing in all our films that had been grown at base pressure. The appearance of FeO in these films occurred for all three oxide precursors as is clear from the XAFS shown in Figure 2a and was a consequence of the low oxygen pressure and not dependent on FeO being in one of the targets. Figure 8a shows the change of the magnetisation measured at room temperature as a function of the oxygen pressure for all of the films. In this case, the amount of oxygen in the target and the PLD chamber produced effects of similar magnitude. It is clear that the largest magnetisation in films without metallic iron was made from a

target that had been produced using FeO. This is interesting because the precursor of choice had been assumed to be $Fe_2O_3$ which was found to be the worst performing precursor in this study.

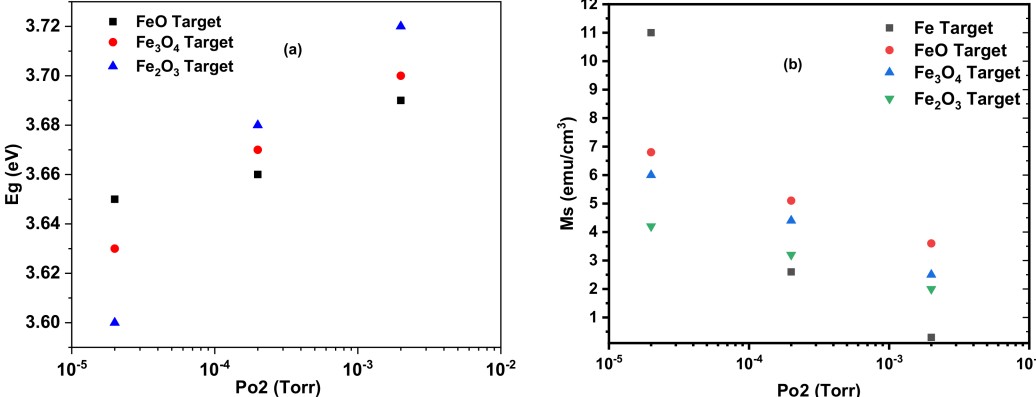

**Figure 8.** The dependence of (**a**) the band gap and (**b**) the saturation magnetisation on the oxygen pressure.

The results of this study are summarised in Figure 8 and Table 1. These show that as the level of oxygen in the PLD chamber is increased the energy of the band edge increases as shown in Figure 8a and the saturation magnetisation reduces as shown in Figure 8b. These results are in agreement with earlier work that deduced that the magnetisation in $(In_{1-x}Fe_x)_2O_3$ is due to oxygen vacancies and the compensating $Fe^{2+}$ ions that are removed by the addition of oxygen [15].

The measurements described here give a coherent account of PLD films made with iron oxide precursors and ablated in different oxygen pressures. The measurements of the lattice constant, XANES and EXAFS spectra, band gap and magnetic measurements were combined to give a clear description of these films. The results fit the general pattern that the oxygen in the target, generated from the precursor, had a similar effect as adding oxygen to the PLD growth chamber. The films grown at base pressure contained some metallic iron as indicated by X-ray absorption and larger coercive fields at 5 K, however, the density of oxygen vacancies was smaller for the precursors, with more oxygen in the target as indicated by the band gap. At higher oxygen pressures, both the density of isolated oxygen vacancies and the density of $Fe^{2+}$ ions were reduced with corresponding drops in the magnetisation, more details are in [38]. The magnetism of the films grown with metallic Fe decreased more rapidly as oxygen was added to the growth chamber. More work should be done on this interesting system.

It has been customary to fabricate targets using $Fe_2O_3$ mixed with $In_2O_3$ because it was assumed that this would naturally be a combination that would be best suited to incorporate the Fe into the $In_2O_3$ lattice. The work done here suggests that FeO would be a better choice because films made with this precursor have a higher magnetisation than those made with $Fe_2O_3$.

**Author Contributions:** Conceptualisation H.B.A. and G.A.G.; Methodology, H.B.A., A.M.A.S., S.M.H., A.M.F. and M.S.A.; Validation, H.B.A. and G.A.G.; Investigation, H.B.A., A.M.A.S., F.-X.J., S.M.H. and K.Y.M.; Resources, H.B.A., G.A.G., S.M.H. and M.S.A.; Writing—original draft preparation, H.B.A.; Writing—review and editing, G.A.G. and M.S.A.; Supervision, G.A.G., A.M.F.; Project administration, G.A.G. and M.S.A.

**Funding:** This research was funded by Saudi Cultural Attaché-London, UK Engineering and Physical Sciences Research EP/D070406/1, Advanced Photon Source, an Office of Science User Facility operated for the US Department of Energy (DOE) Office of Science by Argonne National Laboratory and was supported by the US DOE under Contract No. DE-AC02-06CH11357 and the Canadian Light Source and its funding partners and King Abdulaziz City for Science and Technology (KACST), Saudi Arabia.

**Acknowledgments:** H.B.A. acknowledges the receipt of a studentship from the Saudi Cultural Attaché-London. The SQUID and MCD measurements were taken on apparatus initially funded by the UK Engineering and Physical Sciences Research EP/D070406/1. This research used resources of the Advanced Photon Source, an Office of Science User Facility operated for the US Department of Energy (DOE) Office of Science by Argonne National Laboratory and was supported by the US DOE under Contract No. DE-AC02-06CH11357 and the Canadian Light

Source and its funding partners. In addition, resources for target preparation provided by King Abdulaziz City for Science and Technology (KACST), Saudi Arabia.

**Conflicts of Interest:** The authors have no conflicts of interest.

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
