# Peer review of "Relevance of the Preparation of the Target for PLD on the Magnetic Properties of Films of Iron-Doped Indium Oxide"

_coatings, doi:10.3390/coatings9060381_

Round 1
Reviewer 1 Report
The paper reports preparation of targets for PLD and characterization of the obtained layers via different techniques (PXRD, XANES, EXAFS, magnetic properties measurements, MCD). The topic seems to important because of indium(III) oxide importance. Therefore, proper methods of film preparation and its controlled doping is crucial. The selected PLD method is often used maybe without appropriate control over different factors or at least such details (e.g. grinding technique, sintering, sintering temperature) are missing in the papers. Therefore, this manuscript concern very important task. However, numerous mistakes make it impossible to accept in the current form and require major revision.
Abstract: it is very general (too general). It should be more stand-alone part of the paper presenting the most important findings. Half of the Abstract relates somehow to introduction. Please, rewrite this part focusing more on your achievements.
The most important advantage of this manuscript is related to used variety of techniques, quite often not common. However, the most important doubts concern the Experimental part which is well written with details missing in many other papers for films preparation (it proves that the Authors are experts in this field) whereas characterization techniques are not described. Therefore, it is difficult to follow the experiments performed and results without such details cannot be properly evaluated. Please, complete this part giving details(!!!) for equipment and measurement conditions, e.g. for XANES and EXAFS experiments: beamline setup, detection mode (TEY, fluorescence, absorption), pressure, temperature, how many data collections was averaged, normalization method, software details). Similar description should be completed for all other methods.
Line 51: “using alternate targets of targets…” I do not understand. Please, rewrite.
Line 99: “Although In2O3 can in three…” can what?
Line 116: “Previous work…” the citation is missing.
Line 139: “Appropriate weights of one of the precursors” which, more details, range of weights (if exact values are impossible to present – compare line 147-148).
Line 156-157: “oxygen pressures were used during ablation: base pressure, base pressure, 2×10-5 Torr, and oxygen pressures of 2×10-4Torr and 2×10-3Torr.” I do not understand. Something is redundant or something is missing. Do you mean: “oxygen pressures were used during ablation: base pressure (2×10-5 Torr,) and oxygen pressures of 2×10-4Torr and 2×10-3Torr”?
Line 187: “at ~7017eV (marked with an arrow) that is at a lower energy that that for Wüstite” rather at 7117 eV. “That” is twice (why).
Line 212-222: You observe an intense peak below 2 A. Is it related to Fe-O distances? You observe also many other maxima (in my opinion intense peaks are limited to 7 A). Could you comment their occurrence? For Fig. 3b you mentioned that “the atomic arrangement in the films is very close to that expected for Fe ions substituted on the In sites and distinct from that of hematite”. However, it seems true also for the base pressure - hematite peak at 2.1 A is distinct and all other spectra (apart form substitutional) are very similar with peaks in the same positions. Could you comment/explain. In line 220 you write “the peaks at larger distances”. Where do you think such distinction occurred and what are they related to (e.g. vacancies, thermal displacement, some structural changes)?
Line 234: “in figure 4 and The values” should be “in figure 4 and the values”
Line 235: “are summarized in table 1.” maybe “are summarized in Table 1.”
Line 243: “the Fe ions that were in a metallic environment ~ 12%” and experiment data as well as more detailed discussion is missing. Compare line 206: this value is overestimated according to Authors or not? It definitely requires some comment because data are quoted several times in this paper. Therefore, it seems important to unequivocally comment XANES/EXAFS results.
Line 270-271: “from the Fe metal target at base pressure was not a high as those grown from the oxide targets grown at base pressure” but coercive field at 5K for Fe2O3 is similar (150 vs. 153).
Line 312-314: In line 314 you quote pressure of 2×10-5Torr? Is it correct? In my opinion it should be 2×10-4Torr or2×10-3Torr.
Line 335: “the percentage of the volume occupied by metal are small, of the order of 0.5% or less.)” you wrote about 12% addition of metal – 0.5% addition cannot be detectable at PXRD but for 12% I would expect at least weak signal for the most intense peaks. Could you comment?
Fig. 8b: it is not log plot according to data (rather PO2)!
Author Response
Our responses to the Open Review 1
The paper reports preparation of targets for PLD and characterization of the obtained layers via different techniques (PXRD, XANES, EXAFS, magnetic properties measurements, MCD). The topic seems to important because of indium(III) oxide importance. Therefore, proper methods of film preparation and its controlled doping is crucial. The selected PLD method is often used maybe without appropriate control over different factors or at least such details (e.g. grinding technique, sintering, sintering temperature) are missing in the papers. Therefore, this manuscript concern very important task. However, numerous mistakes make it impossible to accept in the current form and require major revision.
We appreciate these comments on the value of our work and are grateful for the very detailed comments that follow. We have improved the English and presentation considerably as a result of these comments. Where substantial changes have been made we have marked them in red. We are very grateful to this referee in pointing the ways that would really clarify our presentation of this work.
Abstract: it is very general (too general). It should be more stand-alone part of the paper presenting the most important findings. Half of the Abstract relates somehow to introduction. Please, rewrite this part focusing more on your achievements.
The abstract was been rewritten
The most important advantage of this manuscript is related to used variety of techniques, quite often not common. However, the most important doubts concern the Experimental part which is well written with details missing in many other papers for films preparation (it proves that the Authors are experts in this field) whereas characterization techniques are not described. Therefore, it is difficult to follow the experiments performed and results without such details cannot be properly evaluated. Please, complete this part giving details(!!!) for equipment and measurement conditions, e.g. for XANES and EXAFS experiments: beamline setup, detection mode (TEY, fluorescence, absorption), pressure, temperature, how many data collections was averaged, normalization method, software details). Similar description should be completed for all other methods.
We have included more details for the XANES and EXAFS and also for the XRD and magnetisation measurements (these are all indicated in red).
Line 51: “using alternate targets of targets…” I do not understand. Please, rewrite.
So sorry – it was bad English and has been corrected
Line 99: “Although In2O3 can in three…” can what?
Sorry again it was sloppy English and has been corrected
Line 116: “Previous work…” the citation is missing.
The references are now included
Line 139: “Appropriate weights of one of the precursors” which, more details, range of weights (if exact values are impossible to present – compare line 147-148).
This was an interesting comment as, to our knowledge no other author has ever given this detail before. However it is a procedure that needs to be explained carefully to new workers in the field. We have welcomed the opportunity to include this amount of detail.
Line 156-157: “oxygen pressures were used during ablation: base pressure, base pressure, 2×10-5 Torr, and oxygen pressures of 2×10-4Torr and 2×10-3Torr.” I do not understand. Something is redundant or something is missing. Do you mean: “oxygen pressures were used during ablation: base pressure (2×10-5 Torr,) and oxygen pressures of 2×10-4Torr and 2×10-3Torr”?
Sorry again it was sloppy English and has been corrected
Line 187: “at ~7017eV (marked with an arrow) that is at a lower energy that that for Wüstite” rather at 7117 eV. “That” is twice (why).
Again a language slip that has been corrected
Line 212-222: You observe an intense peak below 2 A. Is it related to Fe-O distances? You observe also many other maxima (in my opinion intense peaks are limited to 7 A). Could you comment their occurrence? For Fig. 3b you mentioned that “the atomic arrangement in the films is very close to that expected for Fe ions substituted on the In sites and distinct from that of hematite”. However, it seems true also for the base pressure - hematite peak at 2.1 A is distinct and all other spectra (apart form substitutional) are very similar with peaks in the same positions. Could you comment/explain. In line 220 you write “the peaks at larger distances”. Where do you think such distinction occurred and what are they related to (e.g. vacancies, thermal displacement, some structural changes)?
We thank you for the careful analysis you gave to this aspect of our work. We have rewritten this section with many more details
Line 234: “in figure 4 and The values” should be “in figure 4 and the values”
This was corrected
Line 235: “are summarized in table 1.” maybe “are summarized in Table 1.”
Yes, it is corrected
Line 243: “the Fe ions that were in a metallic environment ~ 12%” and experiment data as well as more detailed discussion is missing. Compare line 206: this value is overestimated according to Authors or not? It definitely requires some comment because data are quoted several times in this paper. Therefore, it seems important to unequivocally comment XANES/EXAFS results.
This is a subtle and important point that is now explained in the section of the x-ray absorption and again when discussing the magnetisation. The percentage of Fe is 5% that of the In. The x-ray absorption data shows that ~12% of these Fe ions are in a metallic environment. This means that only a very small percentage of the whole sample is metallic Fe it is approximately 12% of the 5% ie ~0.6%
Line 270-271: “from the Fe metal target at base pressure was not a high as those grown from the oxide targets grown at base pressure” but coercive field at 5K for Fe2O3 is similar (150 vs. 153).
Thanks. This has been rewritten to make it clear that we thought it useful to compare the results with Fe in the target with those made with FeO. We think the amount of Fe metal would be relatively high for both these films.
Line 312-314: In line 314 you quote pressure of 2×10-5Torr? Is it correct? In my opinion it should be 2×10-4Torr or2×10-3Torr.
Thanks again of course it should have been 2×10-3Torr
Line 335: “the percentage of the volume occupied by metal are small, of the order of 0.5% or less.)” you wrote about 12% addition of metal – 0.5% addition cannot be detectable at PXRD but for 12% I would expect at least weak signal for the most intense peaks. Could you comment?
See comment above. The percentage is actually ~0.6%.
Fig. 8b: it is not log plot according to data (rather PO2)!
True!
We were very grateful for these suggestions because they have led to the manuscript being much improved.
Reviewer 2 Report
Coatings-508202
Title: Relevance of the preparation of the target for PLD on the magnetic properties of films of iron doped indium oxide
The authors present the properties of thin films of iron doped indium oxide grown by PLD. Structural, optical and magnetic properties are exhaustively presented. Three different precursors are used for target preparation: FeO, Fe3O4 and Fe2O3. Also, the influence of oxygen pressure during PLD experiments on the magnetization, optical properties and lattice constants is investigated.
The authors continue their work on the topic. The manuscript is well written, the state-of the art is well addressed; the details of the work and the discussions are convincing.
Minor issues:
1. Please introduce the deposition parameters: laser fluence, number of pulses, target-substrate distance etc.
2. Please rephrase the sentence at lines 78-79-80.
3. Line 187: I think the correct value is 7117 eV.
In my opinion, the topic of the paper makes the work useful and appropriate for Coatings.
Author Response
Our responses to the Open Review 2
The authors present the properties of thin films of iron doped indium oxide grown by PLD. Structural, optical and magnetic properties are exhaustively presented. Three different precursors are used for target preparation: FeO, Fe3O4 and Fe2O3. Also, the influence of oxygen pressure during PLD experiments on the magnetization, optical properties and lattice constants is investigated.
The authors continue their work on the topic. The manuscript is well written, the state-of the art is well addressed; the details of the work and the discussions are convincing.
In my opinion, the topic of the paper makes the work useful and appropriate for Coatings.
We thank the second referee for these supportive coments
Minor issues:
1. Please introduce the deposition parameters: laser fluence, number of pulses, target-substrate distance etc.
Yes, these details were missing and have been added
2. Please rephrase the sentence at lines 78-79-80.
This has been done
3. Line 187: I think the correct value is 7117 eV.
This discussion has been extended and clarified
Round 2
Reviewer 1 Report
I am glad to see this manuscript significantly improved especially by supplying of additional information to Experimental part. Due to that this work is repeatable what should be one of science goals. The variety of used methods makes this paper very interesting and new experimental work details show that this work was properly designed and performed. Now I can accept this paper in the (almost) current form - I would like to ask for verification that (line 204) "...an absorption at ~7017 eV...". It seems that this red arrow points rather 7117 eV!